# Wafer-scale nanofabrication of telecom single-photon emitters in silicon

Michael Hollenbach [1,2] ✉, Nico Klingner[1], Nagesh S. Jagtap[1,2], Lothar Bischoff[1], Ciarán Fowley[1], Ulrich Kentsch[1], Gregor Hlawacek [1], Artur Erbe [1], Nikolay V. Abrosimov[3], Manfred Helm[1,2], Yonder Berencén [1] ✉ & Georgy V. Astakhov [1] ✉

A highly promising route to scale millions of qubits is to use quantum photonic integrated circuits (PICs), where deterministic photon sources, reconfigurable optical elements, and single-photon detectors are monolithically integrated on the same silicon chip. The isolation of single-photon emitters, such as the G centers and W centers, in the optical telecommunication O-band, has recently been realized in silicon. In all previous cases, however, single-photon emitters were created uncontrollably in random locations, preventing their scalability. Here, we report the controllable fabrication of single G and W centers in silicon wafers using focused ion beams (FIB) with high probability. We also implement a scalable, broad-beam implantation protocol compatible with the complementary-metal-oxide-semiconductor (CMOS) technology to fabricate single telecom emitters at desired positions on the nanoscale. Our findings unlock a clear and easily exploitable pathway for industrial-scale photonic quantum processors with technology nodes below 100 nm.

Quantum technologies based on the generation and state manipulation of single photons enable demanding applications[1,2]. A prime example of this is linear optical quantum computation using boson sampling, which requires only single photons and linear optical components[3–5]. The front-runner demonstration is Gaussian boson sampling with 50 single-mode squeezed states[6]. A general-purpose photonic quantum processor can be built using fusing, cluster states, and nonlinear units[7,8]. The latter can be implemented through photon scattering by a two-level quantum system (i.e., a single-photon emitter) coupled to an optical cavity. State of the art for deterministic single-photon sources corresponds to boson sampling with 20 photons using quantum dots (QDs)[9]. To ensure indistinguishability, the same QD routes several photons into a delay line. Delay lines up to 27 m can be realized on a single silicon chip[10], which allows the interference of about 100 deterministic photons. However, the scalability of millions of qubits is not realistic with this approach.

Deterministic single-photon sources monolithically integrated with silicon quantum PIC represent a new tool in quantum photonics[11], complementing heralded probabilistic sources[12] and offering very-large-scale integration (VLSI)[13]. The strategic, long-term goal is the implementation of a photonic quantum processor compatible with present-day silicon technology. Most of the necessary components for cryogenic quantum PICs are available nowadays, including superconducting single-photon detectors[14], delay lines[10], modulators[15], and phase shifters[16]. The practical implementation of this concept has been largely hampered by the lack of controllable fabrication of single-photon emitters in silicon[11,17].

Recently, a broad variety of single-photon emitters have been isolated in commercial silicon-on-insulator (SOI) wafers, including G centers[11,17], W centers[18], T centers[19], some other unidentified damage centers[20], and erbium dopants[21].

Particularly single G centers are carbon-related color centers emitting in the telecom O-band[11,17]. The atomic configuration of the G

[1]Helmholtz-Zentrum Dresden-Rossendorf, Institute of Ion Beam Physics and Materials Research, 01328 Dresden, Germany. [2]Technische Universität Dresden, 01062 Dresden, Germany. [3]Leibniz-Institut für Kristallzüchtung (IKZ), 12489 Berlin, Germany. ✉e-mail: m.hollenbach@hzdr.de; y.berencen@hzdr.de; g.astakhov@hzdr.de

center (Fig. 1a) has been revised several times. According to the latest density functional theory calculations[22], it consists of two substitutional carbon atoms and one interstitial silicon atom in the configuration $C_s - Si_i - C_s$ distorted from the $\langle 111 \rangle$ bond axis (Fig. 1a). The spectroscopic fingerprint of the G center is a spectrally narrow zero-phonon line (ZPL) at $\lambda_G = 1278$ nm in the photoluminescence (PL) spectrum[23]. Another single-photon emitter in silicon is the W center (Fig. 1a), which is ascribed to a tri-interstitial Si complex $I_3$[18]. Like the aforementioned G center, it also possesses a single dipole emission, which has been shown to be polarized along the $\langle 111 \rangle$ crystal axis, revealing a ZPL at $\lambda_W = 1218$ nm in the PL spectrum[23].

Ensembles of the G and W centers in isotopically purified $^{28}$Si crystals reveal extremely narrow linewidths of their ZPLs exceeding the Fourier limit by a factor of two only, which implies marginal spectral diffusion[24]. This makes the G and W centers very promising candidates for the implementation of spatially separated emitters of indistinguishable photons, where the fine-tuning of the emission wavelength can be implemented through the Stark effect or strain control[25,26].

To date, the protocols for the creation of single-photon emitters in silicon consist of either broad-beam implantation of carbon ions at a low fluence ($\Phi \sim 10^9$ cm$^{-2}$)[11] or medium-fluence implantation ($\Phi \sim 10^{12}$ cm$^{-2}$)[17] followed by rapid thermal annealing (RTA)[17]. In both approaches, the process of creating single-photon emitters is not controllable, resulting in emitters being created at random locations. This poses a major obstacle to the realization of wafer-scale quantum PICs with monolithically integrated and on-demand single-photon sources at desired locations.

Here, we use a focused ion beam (FIB)[27–30] to create single G and W centers with a precision better than 100 nm. This concept is illustrated in Fig. 1a. Confirmed by the PL spectra, we unambiguously find that in the case of carbon-rich Si wafers, the Si implantation results in the preferable formation of G centers (the left side of Fig. 1a). For ultrapure silicon wafers and a larger number of Si ions per implantation spot, interstitial complexes rather than G centers are formed, among which are the optically active W centers (the right side of Fig. 1a). In addition to that, we demonstrate large-scale, CMOS-compatible fabrication of single G and W centers using broad-beam Si implantation through lithographically defined nanoholes[31].

## Results

### Creation of single G centers on the nanoscale

To create G centers in a commercial SOI wafer (IceMOS tech.), we perform FIB implantation with double-charged $Si^{2+}$ ions (Fig. 1a). The residual carbon concentration is estimated to be in the range of $10^{16}$ cm$^{-3}$[11]. The Si ions with a kinetic energy of 40 keV are focused to a spot size of about 50 nm. Using the Stopping and Range of Ions in Matter (SRIM) software[32], we calculate the lateral straggling to be $\pm 25$ nm and the mean implantation depth to be $R_p = 60$ nm. The overall spatial resolution is better than 100 nm, both in-depth and laterally (Supplementary Fig. 1).

We generate a FIB pattern consisting of a frame with a dimension of $200 \times 200$ µm$^2$ and $15 \times 16$ individual spots. The frame is created by implanting Si ions at a fluence $\Phi = 1 \times 10^{11}$ cm$^{-2}$. The average number of implanted Si ions per spot is the same in each row and increases logarithmically from $\overline{n}_{Si} = 6$ Si ions for row 1 to $\overline{n}_{Si} = 570$ Si ions for row 15. A detailed list of the averaged number of implanted Si ions ($\overline{n}_{Si}$) per spot is given in Supplementary Table 1. We use the chess notation to label each implanted spot.

After creating the FIB pattern, the samples are measured in a home-built confocal scanning microscope at $T = 6.3$ K under a continuous wave (CW) laser excitation at 637 nm (Supplementary Fig. 2).

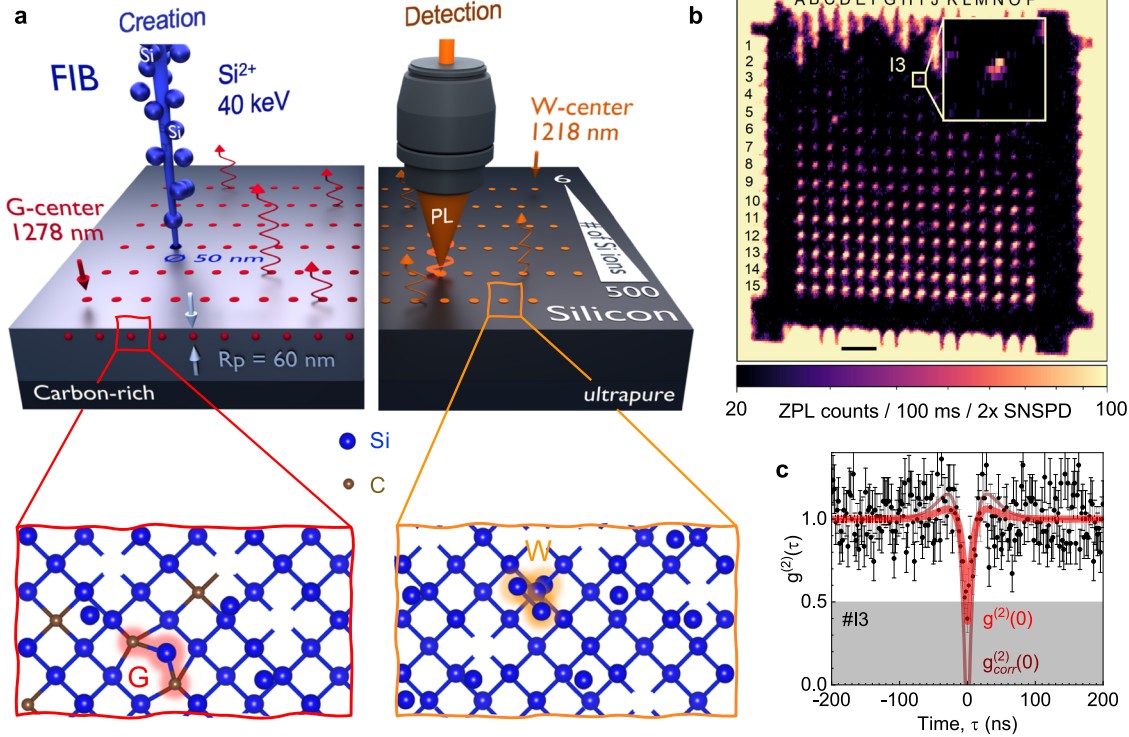

**Fig. 1 | Creation and detection of single G and W centers in silicon. a** Schematic of FIB implantation with $Si^{2+}$ ions and PL collection from single centers. The kinetic energy of 40 keV corresponds to an average implantation depth $R_p = 60$ nm. Si implantation into a carbon-rich and an ultrapure silicon wafer results in the formation of the G and W centers, respectively. **b** Confocal ZPL (1278 nm) intensity map of locally created G centers on an SOI wafer. The number of ions per spot increases logarithmically from nominally 6 (row 1) to 570 (row 15). The pattern frame is created with a fluence $\Phi = 1 \times 10^{11}$ cm$^{-2}$. The scale bar is 20 µm. The inset shows photon emission from a single G center. The color scale is different from the main panel to increase visibility. **c** Second-order autocorrelation function $g^{(2)}(\tau)$ obtained with no BG correction (#I3). The red solid line is a fit to Eq. (1), yielding $g^{(2)}(0) = 0.36 \pm 0.06$. The thin solid line is $g^{(2)}_{corr}(\tau)$ calculated according to Eq. (2). The error bars represent standard deviation (SD).

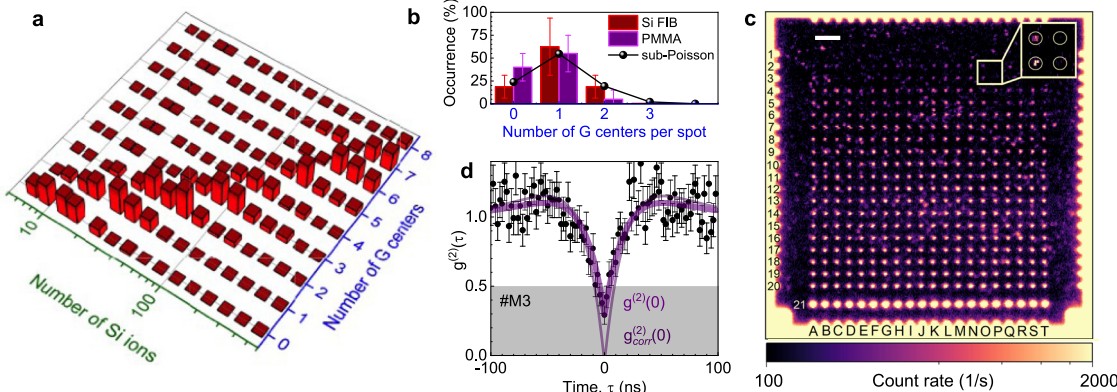

**Fig. 2 | Scalable fabrication of single G centers with sub-100-nm precision in an SOI wafer. a** Statistics histogram representing the probability distribution of the G centers depending on the number of implanted Si ions. **b** The occurrence probability of G centers for FIB implantation (on average 25 Si ions per spot) and Si broad-beam implantation (fluence $1 \times 10^{12}$ cm$^{-2}$) through PMMA holes (nominal diameter 40 nm). The solid line represents the sub-Poisson distribution with $\mu = 4$, as described in the text. **c** Confocal PL intensity map of locally created G centers in an SOI wafer through a PMMA mask using broad-beam Si implantation. The nominal hole diameter increases from 30 nm (row 1) to 400 nm (row 20). The PL is collected using a BP filter $\Delta\lambda = 50$ nm at $\lambda = 1275$ nm. The scale bar is 20 μm. The inset shows four implanted spots with single and no G centers. **d** Second-order autocorrelation function $g^{(2)}(\tau)$ obtained with no BG correction (#M3). The purple solid line is a fit to Eq. (1), yielding $g^{(2)}(0) = 0.22 \pm 0.08$. The thin solid line is $g^{(2)}_{\text{corr}}(\tau)$ calculated according to Eq. (2). The error bars represent SD.

Figure 1b shows a confocal ZPL map. To attenuate the background (BG) contribution, which may be related to the presence of defect states in the bandgap, we use a long pass (LP) filter ($\lambda > 1250$ nm) in combination with a narrow bandpass (BP) filter ($\Delta\lambda = 1$ nm) whose central wavelength coincides with the ZPL of the G center $\lambda_G = 1278$ nm.

To determine the number of G centers in the implanted spots, we measured the second-order autocorrelation function $g^{(2)}(\tau)$ using Hanbury–Brown–Twiss interferometry (Supplementary Fig. 2). The collected photons are coupled to a single-mode fiber and split with a 50/50 fiber beam splitter. The photons are then detected with two superconducting-nanowire single-photon detectors (SNSPDs) with an efficiency >90% in the telecom O-band. The photon detection statistics are recorded with a time-tagging device. An example of such a second-order autocorrelation function from spot #I3 is shown in Fig. 1c with no BG corrections. It is fitted[33]

$$g^{(2)}(\tau) = \frac{N-1}{N} + \frac{1}{N}\left[1 - (1+a)e^{-\left|\frac{\tau}{\tau_1}\right|} + ae^{-\left|\frac{\tau}{\tau_2}\right|}\right]. \quad (1)$$

Here, $N$ corresponds to the number of single-photon emitters. The fit to Eq. (1) yields $g^{(2)}(0) = 0.36 \pm 0.08 < 0.5$ ($N < 2$). From the best fit, we obtain the characteristic antibunching time $\tau_1 \approx 10$ ns. Because of nearly negligible bunching in Fig. 1c, the parameters $\tau_2 \gtrsim \tau_1$ and $a > 0$ cannot be determined with reasonable accuracy.

To increase the photon count rate and consequently decrease the recording time of $g^{(2)}(\tau)$ in Fig. 1c, we use a BP filter with $\Delta\lambda = 50$ nm at $\lambda = 1275$ nm instead of the narrow-band filter as in Fig. 1b. This results in an additional BG contribution to the signal. The autocorrelation function can be corrected due to the presence of the BG as[34]

$$g^{(2)}_{\text{corr}}(\tau) = \frac{g^{(2)}(\tau) - (1 - \rho^2)}{\rho^2}. \quad (2)$$

The constant factor $\rho = (I - B)/I$ takes into account the count rate from an implanted spot ($I$) and the BG, i.e., the count rate from the location in the immediate surrounding the implanted spots ($B$). According to the recent theoretical analysis, the single photon nature of the emission is unambiguously confirmed if the second-order autocorrelation function is zero after the BG and time-jitter corrections[35,36]. The correction due to time jitter (40 ps for the SNSPDs and 14 ps for the time-tagging device) is negligible in Fig. 1c, as it is by more than two orders of magnitude shorter than the $\tau_1$ time. After the BG correction to

Eq. (2), we obtain $g^{(2)}_{\text{corr}}(0) \approx 0$ for spot #I3 (thin solid line in Fig. 1c), which unambiguously points to a single G center ($N = 1$). Remarkably, this G center demonstrates stable operation over hours, with no indication of instability of either the ZPL intensity or the spectrally integrated photon count rate (Supplementary Fig. 3). We note that the spectral resolution is limited by our spectrometer and therefore no conclusions about the ZPL spectral stability can be made. Using this approach, we determine the number of single G centers in other implanted spots.

## Fabrication statistics

The emission from single G centers is linearly polarized and equivalently distributed across four subgroups in the (001) plane[17]. As the excitation energy (1.9 eV) is far above the Si bandgap (1.1 eV), the PL is expected to be independent of the weak elliptical polarization of the excitation. As we collect PL without linear polarizers, we assume the same detection efficiency for all four possible dipole orientations. We assume that the count rate scales linearly with the number of color centers[31] per implantation spot. To estimate an average count rate from the single G center, we use

$$I_G = \frac{\sum_i (I_i - B)}{\sum_i N_i}. \quad (3)$$

Here, $I_i$ is the count rate at the spot $i$ in Fig. 1b obtained from a Gaussian fit (Supplementary Fig. 3) and $N_i$ is the number of G centers established from the BG-corrected autocorrelation function following Eq. (2). We then estimate the number of the G centers in all implanted spots as $N_i = \text{round}\left[(I_i - B)/I_G\right]$. For instance, all spots with a count rate in the range from $0.5I_G$ to $1.5I_G$ are ascribed to single G centers.

Figure 2a summarizes the statistical distribution of the number of G centers ($N$) depending on the average number of implanted Si ions ($\bar{n}_{\text{Si}}$). The mean value of $N$ increases with $\bar{n}_{\text{Si}}$ following a sublinear dependence as expected[11]. The reason is that higher implantation fluence leads to higher crystal damage and, consequently, to a decrease in available crystallographic sites suitable for the formation of G centers. According to the statistics histogram of Fig. 2a, the optimal number of Si ions required to create a single G center is $\bar{n}_{\text{Si}} = 25$ (row 5). The occurrence probability for a different number of G centers, in this case, is presented in Fig. 2b (the red histogram data Si FIB). The probability to create a single G center is as high as $(62 \pm 31)\%$, while there is a lower but nonzero probability of creating multiple or no G centers at the implantation spots. Though within the error bars, the

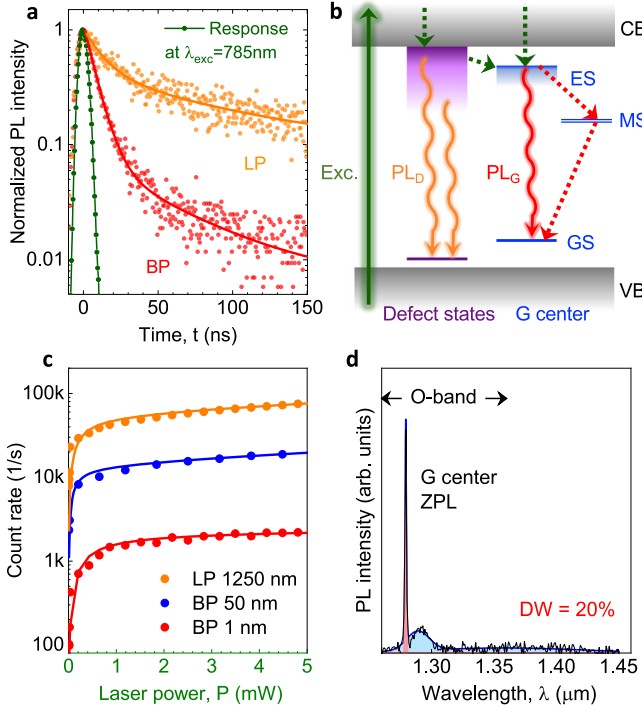

**Fig. 3 | Photoexcitation of G centers. a** PL decay of the locally created G centers obtained with an LP filter $\lambda > 1250$ nm (orange) and a BP filter $\Delta\lambda = 1$ nm at $\lambda_G = 1278$ nm, corresponding to the ZPL of the G center (red). The solid lines are fits to a bi-exponential decay. The excitation laser pulse at $\lambda_{exc} = 785$ nm is also shown for comparison (green). **b** Schematic representation of the excitation and recombination processes of the G center (PL$_G$) in the vicinity of bandgap defect states (PL$_D$). **c** Count rate of a single G center as a function of the excitation power in the presence of BG obtained with different optical filters: LP filter $\lambda > 1250$ nm (orange), BP filter $\Delta\lambda = 50$ nm at $\lambda = 1275$ nm (blue) and BP filter $\Delta\lambda = 1$ nm at $\lambda_G = 1278$ nm (red). The solid lines are fits to Eq. (4). **d** PL spectrum of a single G center, obtained at $P = 100$ μW. A multi-Gauss fit over the ZPL and PSB's (blue solid line) yields a Debye–Waller factor DW = 20%.

distributions of Fig. 2a, b can be described by the Poisson function, there is a strong indication that the experimental data deviate from it. Considering that the G center is a composite defect consisting of three atoms, we can reproduce the sub-Poisson statistics shown by the solid line in Fig. 2b (Supplementary Fig. 4). The real formation process of the G centers is much more complex than in our simplified model based on a multi-step Si implantation process (Supplemental Material) and beyond the scope of this work.

To analyze the BG contribution, we perform time-resolved PL measurements with an LP and a narrow BP filter (Fig. 3a). The PL spectrum, together with the filter transmission wavelengths, is shown in Supplementary Fig. 5. The PL decay is fitted to a bi-exponential function. The fast PL decay with a time constant of about 10 ns dominates when the narrow BP filter is tuned to the ZPL[37]. Therefore, this is associated with the G center. For the spectrally integrated decay, i.e., with the LP filter only, there is a slow contribution with a time constant of about 70 ns. This is ascribed to the presence of defect states in the bandgap, which are created during the fabrication of the SOI wafer. The excitation and recombination processes involving the defect states and G centers are schematically presented in Fig. 3b. This explanation is also confirmed by the excitation power ($P$) dependence of the PL count rate ($I$) for three different filter configurations (Fig. 3c). It is fitted to

$$I(P) = \frac{I_G(\lambda)}{1 + P_0/P} + S_D(\lambda)P, \qquad (4)$$

where $I_G(\lambda)$ is the saturation count rate and $S_D(\lambda)$ is a spectrally-dependent slope describing the BG contribution. The fit of $I(P)$ integrated over the ZPL and the phonon sideband (PSB), i.e., with a BP filter 50 nm, gives $I_G = 13 \times 10^3$ counts per second. We find the saturation excitation power for this case $P_0 = 110$ μW, which can be reduced using an optimum excitation wavelength according to the PL excitation spectrum[17,37] (Supplementary Fig. 5).

### Wafer-scale fabrication of single G centers

To reduce the BG in our commercial SOI wafers, a series of RTA and furnace annealing (FA) experiments were performed (Supplementary Fig. 6). We find that the most efficient reduction is obtained with RTA processing at 1000 °C for 60 s. After optimizing the implantation and annealing parameters, we demonstrate the controllable creation of single G centers using a CMOS-compatible protocol. We first fabricate a PMMA mask with lithographically defined arrays of nanoholes (Supplementary Fig. 1) having different diameters (Supplementary Table 2). Then, we perform a broad-beam implantation with Si ions at a fluence $\Phi = 1 \times 10^{11}$ cm$^{-2}$ and with the same kinetic energy of 40 keV as in the FIB experiments.

A confocal map of the G centers created in $20 \times 20$ nanoholes is depicted in Fig. 2c. The PL count rate is spectrally integrated over the ZPL and PSBs. As an illustration, we show the autocorrelation function recorded at the spot #M3 with no BG correction (thick solid line in Fig. 2d). The fit to Eq. (1) yields $g^{(2)}(0) = 0.22 \pm 0.08$ and after BG correction to Eq. (2) $g^{(2)}_{corr}(0) \approx 0$ (the thin solid line in Fig. 2d), pointing to a single-photon emission. Some other $g^{(2)}(\tau)$ measurements of single G centers at different implanted spots are shown in Supplementary Fig. 7. Based on the $g^{(2)}(\tau)$ measurements and calibrated count rate, we find that more than 50% of the nanoholes with nominal diameters of 35 and 40 nm (rows 2 and 3, respectively) contain single G centers (Fig. 2b).

Figure 3d shows a PL spectrum from the spot with a nominal diameter of 40 nm (#M3). It consists of the ZPL at $\lambda_G = 1278$ nm and the PSB with a maximum at around 1290 nm[17]. The Debye–Waller (DW) factor, i.e., the probability of coherently emitting into the ZPL, is an important characteristic of single-photon emitters for their applications in photonic quantum technologies. We find DW = 19 ± 1%. This is the largest value reported to date for individual G centers and is comparable with a DW factor of an ensemble of G centers with an optimized creation protocol[37].

### Creation of single W centers on the nanoscale

Finally, we turn to the controlled creation of W center emitters with the ZPL at $\lambda_W = 1218$ nm. In order to locally write W centers, we use the same procedure as for G centers in SOI, with the difference that the substrates are now ultrapure Si wafers with negligible carbon content (Fig. 1a). After implantation, the sample is annealed at 225 °C for 300 s[24,38]. Figure 4a shows a confocal PL map of this pattern. A 50-nm BP filter at 1225 nm is used to selectively collect the PL emission from the ZPL and the first PSB of W centers. We optically resolve all the implanted spots in row 15 (on average, 570 Si ions per spot) down to only a few implanted spots in row 10 (on average, 113 Si ions per spot).

We show an autocorrelation measurement at a spot irradiated with, on average, 384 Si ions (#I14) with no BG correction (Fig. 4b). The dip at $\tau = 0$ indicates a countable number of W centers ($N \leqslant 5$). We observe a relatively high BG (Supplementary Fig. 8). A possible reason is that we use an established annealing protocol optimized for a dense ensemble of W centers[38], which might not be optimum for the creation of individual W centers. Applying the BG correction procedure of Eq. (2), we obtain $g^{(2)}_{corr}(0) = 0.13^{+0.35}_{-0.13}$, which indicates that, in fact, we have single-photon emission from this spot. To find the power dependence of the photon count rate from a single W center of Fig. 4c, we subtract the BG contribution taken from the non-implanted area between the nearest spots. A fit to Eq. (4) gives $I_W = 3600$ counts per second and

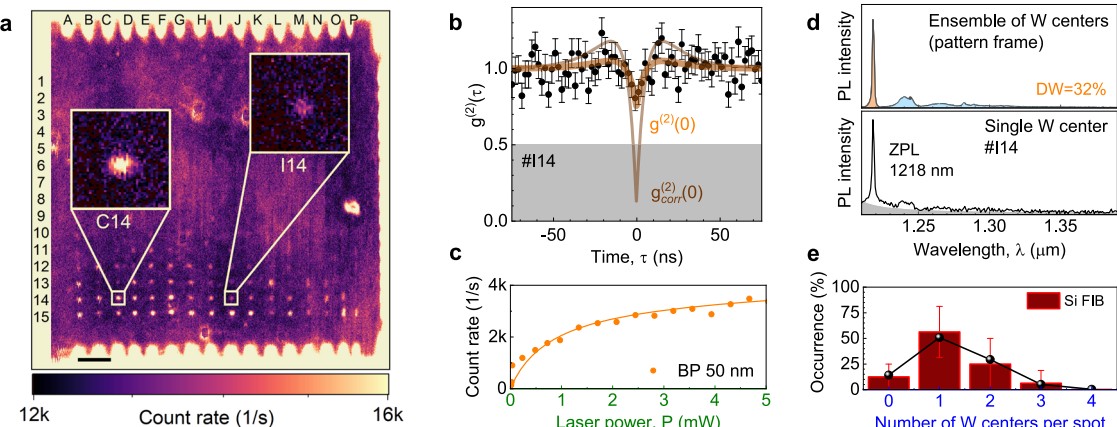

**Fig. 4 | Single W centers in ultrapure silicon. a** Confocal PL intensity map of locally created W centers. The pattern frame is created with a fluence $\Phi = 1 \times 10^{11}$ cm$^{-2}$. The PL is collected using a BP filter $\Delta\lambda = 50$ nm at $\lambda = 1225$ nm. The scale bar is 20 μm. The insets show the PL from two W centers (#C14) and a single W center (#I14). **b** Second-order autocorrelation function $g^{(2)}(\tau)$ obtained at the spot #I14 with no BG correction. The BG correction (thin solid line) gives $g^{(2)}_{corr}(0) = 0.13^{+0.35}_{-0.13}$. **c** Count rate of a single W center after BG subtraction as a function of the excitation power, which

is measured with a BP filter $\Delta\lambda = 50$ nm at $\lambda = 1225$ nm. The solid line is a fit to Eq. (4). **d** PL spectrum from the frame and a single W center (#I14). Integration over the ZPL and PSB yields a Debye–Waller factor DW = 32%. The BG contribution is schematically shown by the shaded area. **e** The occurrence probability of W centers for FIB implantation with, on average, 384 Si ions per spot. The solid line represents the sub-Poisson distribution with $\mu = 4.8$, as described in the text. The error bars represent SD.

$P_0 = 810$ μW (Supplementary Fig. 8), which is lower than the saturation count rate of the G centers.

A PL spectrum from a single W center is shown in the lower panel of Fig. 4d, which is similar to the PL spectrum of an ensemble of W centers (upper panel of Fig. 4d). We find a DW = 32%, which is significantly larger than that for the G center. For low excitation powers ($P \ll P_0$), the PL spectrum and photon count rate remain stable over one day of operation (Supplementary Fig. 8). For high excitation powers ($P > P_0$), we observe blinking of the ZPL. The origin of this optical instability is beyond the scope of this work.

Two spots with implantation $\bar{n}_{Si} = 384$ (row 14) show a difference in count rate, after the BG correction, of a factor of two, indicating that one contains a single center (#I14) and one contains two single centers (#C14). This is in agreement with the corrected $g^{(2)}_{corr}(0) = 0.52 \pm 0.15$ indicating two-photon emission (Supplementary Fig. 9). The emission from the W centers is linearly polarized either along the [110] or [1$\bar{1}$0] direction[18] and, similar to the G centers, we assume the same detection efficiency for both directions. Based on the $g^{(2)}(\tau)$ and the photon count rate analysis of the implanted row 14, we find that, in a similar way to Fig. 2b, the creation probability of a single W center is $(56 \pm 28)\%$ (Fig. 4e). Thus, the analysis indicates that the W centers are created with sub-Poisson statistics, as explained in Supplementary Fig. 4.

To demonstrate the wafer-scale fabrication of single W centers, we use broad-beam Si implantation through nanoholes in a PMMA mask. The implantation parameters (Si$^{2+}$ with a kinetic energy of 40 keV, $\Phi = 1 \times 10^{12}$ cm$^{-2}$) and mask design (Supplementary Table 2) are similar to those optimized for the fabrication of single G centers. A confocal map of the W centers created in 20 × 20 nanoholes is depicted in Fig. 5a. The PL count rate is spectrally integrated over the ZPL and the first PSB. Because post-implantation annealing at 225 °C leads to a high BG in Fig. 4a, no annealing is performed. Indeed, the BG in the confocal map of Fig. 5a is significantly lower.

As an illustration, we show the autocorrelation function recorded at spot #C13 (nanohole with a nominal diameter of 300 nm) without BG correction (Fig. 5b). The fit to Eq. (1) yields $g^{(2)}(0) = 0.48 \pm 0.27$. With BG correction to Eq. (2), we obtain $g^{(2)}_{corr}(0) \approx 0$ demonstrating single-photon emission. We note that the absence of annealing also leads to a lower count rate from single W centers because the lattice damage after implantation is not removed. The optimization of the annealing conditions for the local creation of single W centers with a high photon

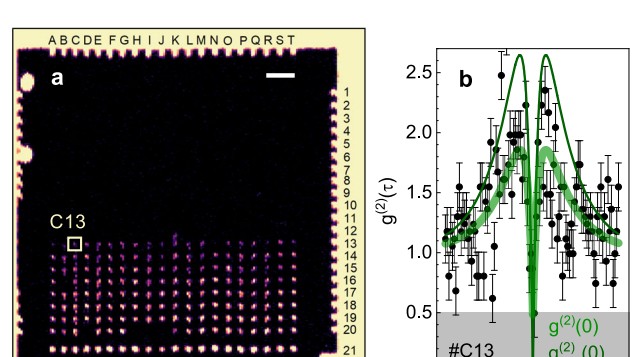

**Fig. 5 | Scalable fabrication of single W centers in ultrapure silicon. a** Confocal PL intensity map of locally created W centers through a PMMA mask using broad-beam Si implantation. The nominal hole diameter increases from 30 nm (row 1) to 400 nm (row 20). The PL is collected using a BP filter $\Delta\lambda = 50$ nm at $\lambda = 1225$ nm. No annealing is performed. The scale bar is 20 μm. **b** Second-order autocorrelation function $g^{(2)}(\tau)$ obtained with no BG correction from the spot #C13. The green solid line is a fit to Eq. (1), yielding $g^{(2)}(0) = 0.48 \pm 0.27$. The thin solid line is $g^{(2)}_{corr}(\tau)$ calculated according to Eq. (2). The error bars represent SD.

emission rate and low BG contribution is a technologically challenging task and is beyond the scope of this work.

## Discussion

In summary, we unambiguously demonstrate the controllable creation of quantum telecom emitters based on single silicon-interstitial- and carbon-related color centers in silicon wafers. These single-photon emitters are created with a spatial resolution better than 100 nm and a probability exceeding 50%. Using broad-beam implantation through lithographically defined nanoholes, we demonstrate the wafer-scale nanofabrication of telecom single-photon emitters compatible with CMOS technology for VLSI. Our results enable the direct realization of quantum PICs with monolithically integrated single-photon sources with electrical control[11]. These findings also provide a route for the quasi-deterministic creation of single G and W centers at desired

locations of photonic structures[39], tunable cavities[40], and SOI waveguides[41]. Furthermore, our approach can potentially be applied for the controllable creation of other color centers in silicon, including T centers with optically-interfaced spins[19].

## Methods

### Samples

Two different sets of p-type silicon wafers are utilized for the experiments. In the case of G centers, we performed our experiments on a commercially available Czochralski (CZ)-grown ⟨110⟩-oriented SOI wafer purchased from IceMOS Technology. This wafer consists of a 12-μm-thick Si device layer separated by a 1-μm-thick silicon dioxide (SiO$_2$) layer from the bulk silicon substrate. The double-side polished 315-μm-thick substrate is cleaved into $5 \times 5$ mm$^2$ pieces. The as-grown concentration of carbon impurities for this type of wafers is specified to be higher than $10^{16}$ cm$^{-3}$ [11]. To decrease the natural BG contribution, we perform either FA or RTA in an N$_2$ atmosphere.

To investigate W centers, we use ⟨100⟩-oriented single-side polished, 525-μm-thick, ultrapure silicon substrates grown by the float zone (FZ) technique. The residual concentration of carbon and oxygen impurities is less than $5 \times 10^{14}$ cm$^{-3}$ and $1 \times 10^{14}$ cm$^{-3}$, respectively, whereas the concentration of boron and phosphorous dopants falls below $7 \times 10^{12}$ cm$^{-3}$. To create the optically active W center, we performed FA at 225 °C for 300 s in an N$_2$ atmosphere following fabrication protocols optimized for an ensemble of W centers[24,38].

### FIB implantation

We used a customized Orsay Physics CANION Z31Mplus FIB system with a liquid metal alloy ion source (LMAIS). The FIB system is equipped with an in-house-fabricated Au$_{82}$Si$_{18}$ ion source, which provides a focused ion beam with a diameter of roughly 50 nm[42]. The small focal spot of the FIB offers fast, flexible, maskless, and spatially resolved targeted positioning of the implanted ions at the nanoscale. Additionally, the system is equipped with a Wien ExB mass filter to block different ion species and charge states emerging from the ion source. The double-charged Si$^{2+}$ ions with a nominal beam current between 1 and 2.5 pA have a kinetic energy of 40 keV (at 20 kV acceleration potential).

For the FIB implantation of single G and W centers, a custom patterning file is created for both the frame and the single dot arrays, respectively. The frame is implanted with a constant fluence $\Phi \sim 10^{11}$ cm$^{-2}$ to intentionally create a dense ensemble of color centers for reference and alignment purposes. For the individual single dot arrays with $15 \times 16$ irradiation spots (vertical and horizontal spacing 10 μm), the number of ions per spot is targeted to be between 6 to 570 with logarithmic incremental steps. The implanted number of Si ions is controlled by the dwell time, such that the desired dose of Si ions is reached.

### Broad-beam implantation

SOI wafers are processed using an RTA at 1000 °C for 3 min under an N$_2$ atmosphere, 15 min of piranha (3 parts H$_2$SO$_4$ : 1 part H$_2$O$_2$) cleaning is performed to remove residual carbon- and oxygen-terminate the sample surface. Prior to resist spin coating, the samples are ultrasonically cleaned in acetone, rinsed in IPA, and blown dry with N$_2$. Next, a layer of positive micro resist (PMMA, 950K A6) with a nominal thickness of $t = 324$ nm is spin-coated on the wafer as an implantation mask. Subsequently, the sample is baked on a hot plate for 5 min at 150 °C. The nanohole patterns, containing $20 \times 20$ of variable diameters $d$ ranging from 30 to 400 nm, were transferred to the photoresist by electron beam lithography (EBL) utilizing a Raith 150TWO system. To tune the number of implanted Si ions through different nanoholes, we vary the nominal nanohole diameter while keeping the EBL dose constant. The overall design, including the lateral 10 μm pitch between all nanoholes, was chosen for comparison and consistency with the irradiation pattern used for the FIB writing. During the EBL process, the following parameters are used: 20 kV acceleration voltage, 0.25 nA current, 30 μm aperture with a base dose of 820 μC · cm$^{-2}$. After the EBL, the PMMA is developed with a mixture of DI-water and isopropyl alcohol (3:7) for 30 s followed by an isopropyl alcohol stopper for 30 s, the samples are then dried with pressurized nitrogen. To create single G centers for VLSI, we use broad-beam implantation with Si$^{2+}$ ions (energy 40 keV) through the micro resist mask with a fluence of $\Phi = 1 \times 10^{12}$ cm$^{-2}$ at $\theta = 7°$ tilt to avoid ion channeling. After the lift-off process, ultrasonication in acetone for 3 min is applied to remove the residuals of PMMA, followed by washing in isopropyl alcohol and blow-drying under a stream of nitrogen gas.

According to SRIM calculations[32], the $R_p$ of 40 keVSi$^{2+}$ in PMMA is ~100 nm. Therefore, ions only reach the substrate through the holes in the mask. To prevent the unwanted creation of other types of emitting color centers, no post-irradiation annealing treatment was performed.

## Data availability

The experimental data are available upon request.

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

## Acknowledgements

We thank Ilona Skorupa for the help with FA, Gabriele Schnabel for piranha, Bernd Scheumann for associated metal depositions during EBL optimization, and Helmut Schultheiss for the assistance with Blender by the preparation of schemtics in Fig. 1a and Supplemenatry Fig. 2. Support from the Ion Beam Center (IBC) at HZDR for ion implantation and Nanofabrication Facilities Rossendorf (NanoFaRo) at IBC is gratefully acknowledged.

## Author contributions

M.Ho., Y.B., and G.V.A. conceived and designed the experiments. M.Ho. performed the single-photon spectroscopy experiments under the supervision of G.V.A. M.Ho., N.K., and L.B. designed the FIB layout. N.K. and L.B. performed FIB implantation and in situ annealing. M.Ho., N.S.J., Y.B., C.F., and G.V.A. designed the PMMA mask. N.S.J. fabricated the PMMA mask. M.Ho., U.K., Y.B., and G.V.A. conceived and performed the broad-beam silicon implantation. C.F. and Y.B. carried out the RTA processing. N.V.A. grew the ultrapure silicon substrates. M.Ho. and G.V.A. wrote the manuscript. All authors, together with G.H., A.E., and M.H. discussed the results and contributed to the manuscript preparation.

## Funding

## Competing interests

The authors declare no competing interest.
