## [Peer Review File · Nature Communications]

REVIEWER COMMENTS

Reviewer #1 (Remarks to the Author):

The Authors present experimental results of controlled positioning of single quantum emitters in silicon. Telecom-wavelength colour centres in silicon have very recently been isolated by several teams, including the Authors, but with creation techniques that lead to random positioning on the substrate. Here, the Authors demonstrate two methods, based on focused ion beam and on nanohole masking, that yield deterministically positioned (< 100 nm) emitters with high yield (≥ 50 %). The two processes for G-centre creation from C-doped silicon are already of high quality and can be readily applied for more complex experiments. The Authors also generate W-centres from high purity silicon, and although it will necessitate further optimisation, they are already able to demonstrate single emitters – yet with strong background luminescence. Their statistical analysis is overall nicely conducted and provides useful insight on the optimal irradiation parameters. This work constitutes an important step towards silicon-based integrated quantum photonics since it can unlock top-down integration of quantum emitters in silicon photonic devices. I support publication in Nature Communications provided that the Authors address the following minor remarks and questions.

1. Have the Authors tried masked broad-beam implantation of W centres? What were/would be the potential issues?

2. In the “Fabrication statistics” section, I agree with the method to infer the number of emitters from combined g^2 and intensity. However, the Authors claim that the count rates from implanted spots show stepwise increases. This is not so clear to me from figure S3a and I have several questions to this regard:

- They did fit the intensity profile by Gaussian functions, but with varying widths. Should there be a stepwise distribution, would it not be of the area rather than of the maximum value?

- I expect the irradiation process to generate randomly oriented G-centre dipoles, also leading to randomly oriented radiation patterns, thus to continuously varying count rate as collected from a finite numerical aperture. Why do the Authors expect a discrete intensity distribution?

- The Authors should indicate the excitation polarisation, which also plays a role in the laser-emitter coupling, and therefore on the individual count rate at a given power.

3. Page 4, the Authors should perhaps explain at this point why a sublinear dependence is expected – is it because the C density is fixed and therefore the G-centre density saturates? Is this also the case for W-centres?

4. Fig 2b: If I understand correctly, the sub-Poissonian distribution considered by the Authors in supplementary section 2 and figure S4 is a re-binned Poisson distribution of $\mu = 4$ that is down-sampled by groups of three, corresponding to the number of atoms in the W complex. However, Figure 2b deals with G centres, and it is not clear if the Authors used the same method as for the W centres, and why this should be the case – in particular, given that G centres are created by Si implantations with C atoms already in the lattice. Did the Authors use the same 3-fold down-sampling for G centres? Do they have a qualitative argument for this sub-Poissonian behaviour in the case of G centres?

Reviewer #2 (Remarks to the Author):

In their manuscript, the authors demonstrate the localized creation of single photon sources in silicon through either masked ion implantation or irradiation with a focused ion beam. The authors provide a clean set of measurements demonstrating photoluminescence from generated single G centres and W centres. This is a very timely work as research into silicon-hosted colour centres is on the rise, and this manuscript will certainly be read with interest by many members of the community.

A few points in the manuscript require additional clarification, but once these issues are addressed, I recommend the manuscript for publication.

- In the introduction, the authors note that "Recently, a broad variety of single-photon emitters have been isolated in commercial silicon-on-insulator (SOI) wafers." They then cite a number of papers demonstrating the creation of G and W centres. Citations could be added for works demonstrating the incorporation of erbium (maybe arXiv:2005.01775) and T centres (maybe arXiv:2103.07580) into SOI material.

- At the end of the introduction, the claim is made that "Here, we use a focused ion beam (FIB) to create single G and W centers with nanometer precision." The centres are later shown to be

localized with <100nm precision. The claim in the introduction should be rephrased to match the demonstrated precision.

- The claim made that there is "no indication of instability of...the ZPL" for a single G centre is not convincing in the current format. Asserting stable ZPL emission is a very strong statement that implies there is no spectral diffusion of the optical transition. If this is true, more details need to be added to strengthen this claim. The data shown in Fig. S3d appears to be the source of this claim, but this data does not quote the linewidth of the single G centre ZPL. For the ZPL emission to truly be stable, the measured linewidth needs to be equal to the emitter's lifetime-limited linewidth in each of the scans contributing to Fig. S3d. If this is not the case, then the ZPL could be broadened by spectral diffusion occurring at a timescale faster than the measurement. From the axis ticks in Fig. S3d, the ZPL does not appear to be lifetime-limited. Moreover, from the caption it appears a 1nm bandpass filter was in place for this measurement, and to the eye, it looks like the ZPL linewidth is comparable to the bandwidth of this filter. The data in Fig. S3e certainly demonstrates stability in the integrated photon count rate, but it does not appear to me that stability of the ZPL has been demonstrated.

- The measured value of 20% for the Debye-Waller factor for the G centre is said to disagree with previous reported values. How much does it deviate from previous values and is such a deviation expected? DW factors can be tricky to measure if spectra are not properly corrected for the wavelength dependence of the collection path and detector. Has such a correction been applied to this result or is there data demonstrating a correction is not needed?

Reviewer #3 (Remarks to the Author):

The paper contains important novel results on scalable fabrication of near-infrared emitters in silicon, especially, the G-centers and W-centers. The paper claims that they could prove fabrication of single photon emitters compatible with CMOS technology. The topic is indeed a potential subject for Nature Communications.

The main problem of the paper is about the goodness of the measurement of the single photon emitters that is the key focus of the entire paper. In the paper, the single photon nature of the emission is defined as $g_2(0) < 0.5$, however, it has been shown recently that this definition is misleading because, in practice, $g_2(0) < 0.5$ condition does not absolutely guarantee single photon emission because two-three emitters may also produce $g_2(0) \sim 0.4$, for instance. Recent discussions can be found in Refs. arXiv:2111.01252 and arXiv:2203.11859. This issue should be resolved tranquillizingly before considering for recommendation for publication this paper in any form.

We thank all the Reviewers for their useful comments and suggestions, which helps us to improve our manuscript. Here, we provide a point-by-point response to the Reviewers' comments. The main changes are highlighted in the revised manuscript file. To address all the issues raised by the Reviewers, we have performed new experiments (Fig. 5) and done additional analysis. Now, we demonstrate focused- and broad-beam protocols for the controllable creation of both single G and W centers. Therefore, we believe that our manuscript can be published in its present form.

Reviewer #1

"I support publication in Nature Communications provided that the Authors address the following minor remarks and questions."

We thank the Reviewer for the support of the publication of our manuscript.

"Have the Authors tried masked broad-beam implantation of W centres? What were/would be the potential issues?"

We have performed masked, broad-beam implantation of the W centers in the revised version of the manuscript. It is presented in Fig. 5.

"In the "Fabrication statistics" section, I agree with the method to infer the number of emitters from combined $g^{(2)}$ and intensity. However, the Authors claim that the count rates from implanted spots show stepwise increases. This is not so clear to me from figure S3a and I have several questions to this regard:"

We have removed this claim but we assume that the count rate scales linearly with the number of color centers per implantation spot. The number of defects is calculated based on the count rate within a certain interval. For instance, we set this interval for single G centers from $0.5 I_G$ to $1.5 I_G$, where I_G is the average count rate for single G centers confirmed from the independent $g^{(2)}$ measurements.

"- They did fit the intensity profile by Gaussian functions, but with varying widths. Should there be a stepwise distribution, would it not be of the area rather than of the maximum value?"

A small variation of the intensity spatial profile is caused by our nanopositioner because we use a fast scanning mode in the open-loop configuration, when the step size slightly varies. The calibration of the nanopositioner step size is then performed using the pattern frame with known dimensions. It allowed us to reduce the scanning time without losing generality. To show that it does not change our conclusions, we fit now the data in Fig. S3a using fixed width.

"- I expect the irradiation process to generate randomly oriented G-centre dipoles, also leading to randomly oriented radiation patterns, thus to continuously varying count rate as collected from a finite numerical aperture. Why do the Authors expect a discrete intensity distribution?"

The emission from the G centers is linearly polarized and equivalently distributed across four subgroups in the (001) plane [Redjem et al., Nat. Electron. 3, 738 (2020)]. The emission from the W centers is also linearly polarized across two subgroups with orthogonal polarizations in the (001) plane [Baron et al., ACS Photon. 9, 2337 (2022)]. As we collect photoluminescence from the (001) plane without linear polarizers, the photon count rate should be the same for all possible dipole orientations of the G and W centers. We have added this discussion in the main text (section "Fabrication statistics").

"- The Authors should indicate the excitation polarisation, which also plays a role in the laser-emitter coupling, and therefore on the individual count rate at a given power."

We use single-mode fiber-pigtailed laser diodes for the excitation. This fiber does not maintain the linear polarization of the laser. Though there is a small remaining elliptical polarization of the

laser excitation after passing the fiber, it does not contribute to the PL polarization because the excitation energy (1.9 eV) is far above the Si bandgap (1.1 eV). Therefore, we excite electrons directly from the valence band to the conduction band. The electrons relax fast to the bottom of the conduction band and the information about excitation polarization is lost. The electrons are then captured by the excited state of the G and W centers followed by radiative recombination. We now indicate in the main text (section "Fabrication statistics") that the excitation is weakly elliptically polarized.

"Page 4, the Authors should perhaps explain at this point why a sublinear dependence is expected – is it because the C density is fixed and therefore the G-centre density saturates? Is this also the case for W-centres?"

The reason is that higher implantation fluence leads to higher crystal damage and, consequently, to a decrease of available crystallographic sites suitable for the formation of G and W centers. We have added this explanation in the main text.

"Fig 2b: If I understand correctly, the sub-Poissonian distribution considered by the Authors in supplementary section 2 and figure S4 is a re-binned Poisson distribution of $\mu = 4$ that is down-sampled by groups of three, corresponding to the number of atoms in the W complex. However, Figure 2b deals with G centres, and it is not clear if the Authors used the same method as for the W centres, and why this should be the case – in particular, given that G centres are created by Si implantations with C atoms already in the lattice. Did the Authors used the same 3-fold down-sampling for G centres? Do they have a qualitative argument for this sub-Poissonian behaviour in the case of G centres?"

Similar arguments can be applied for the formation of the G center, consisting of two C atoms and one Si atom. Because the probability to find two C atoms in the neighboring crystal sites is vanishingly small, the formation of a single G center can be considered as a multi-step implantation process: (1) implanted silicon ion kicks out a silicon atom from a lattice site adjacent to a carbon substitutional and forms a vacancy, (2) implanted silicon ion kicks out a carbon atom from one substitutional into the vacancy and forms a carbon pair, (3) implanted silicon ion creates silicon interstitial adjacent to the carbon substitutional pair and forms a G center. The real formation processes of the G (and W) centers under Si implantation are much more complex than in our simplified consideration and beyond the scope of this work. Nevertheless, our simplified model based on a multi-step Si implantation process can qualitatively explain sub-Poisson statistics. This qualitative explanation is now given in the Supplemental Material (page 4) and briefly discussed in the main text.

Reviewer #2

"This is a very timely work as research into silicon-hosted colour centres is on the rise, and this manuscript will certainly be read with interest by many members of the community."

We are glad to know that our research is timely and of potential interest for many members of the community.

"- In the introduction, the authors note that "Recently, a broad variety of single-photon emitters have been isolated in commercial silicon-on-insulator (SOI) wafers." They then cite a number of papers demonstrating the creation of G and W centres. Citations could be added for works demonstrating the incorporation of erbium (maybe arXiv:2005.01775) and T centres (maybe arXiv:2103.07580) into SOI material."

We now cite these papers in the introduction.

"- At the end of the introduction, the claim is made that "Here, we use a focused ion beam (FIB) to create single G and W centers with nanometer precision." The centres are later shown to be

localized with <100nm precision. The claim in the introduction should be rephrased to match the demonstrated precision.”

We have rephrased it to “with a precision better than 100 nm”.

“- The claim made that there is “no indication of instability of...the ZPL” for a single G centre is not convincing in the current format. Asserting stable ZPL emission is a very strong statement that implies there is no spectral diffusion of the optical transition. If this is true, more details need to be added to strengthen this claim. The data shown in Fig. S3d appears to be the source of this claim, but this data does not quote the linewidth of the single G centre ZPL. For the ZPL emission to truly be stable, the measured linewidth needs to be equal to the emitter’s lifetime-limited linewidth in each of the scans contributing to Fig. S3d. If this is not the case, then the ZPL could be broadened by spectral diffusion occurring at a timescale faster than the measurement. From the axis ticks in Fig. S3d, the ZPL does not appear to be lifetime-limited. Moreover, from the caption it appears a 1nm bandpass filter was in place for this measurement, and to the eye, it looks like the ZPL linewidth is comparable to the bandwidth of this filter. The data in Fig. S3e certainly demonstrates stability in the integrated photon count rate, but it does not appear to me that stability of the ZPL has been demonstrated.”

The Reviewer points out a very important issue. Actually, we are referring to the stability of the ZPL intensity (for instance, that the ZPL does not blink), but not the ZPL spectral stability. The spectral resolution is limited by our spectrometer and therefore no conclusions about the ZPL spectral stability can be made. We now rephrase and clearly indicate this in the revised version of the manuscript to avoid misinterpretation of our claim (the last sentences in the section “Creation of single G centers on the nanoscale”).

“- The measured value of 20% for the Debye-Waller factor for the G centre is said to disagree with previous reported values. How much does it deviate from previous values and is such a deviation expected? DW factors can be tricky to measure if spectra are not properly corrected for the wavelength dependence of the collection path and detector. Has such a correction been applied to this result or is there data demonstrating a correction is not needed?”

The strongest contribution to the wavelength dependence comes from the grating, whose sensitivity drops from 85% at 1300 nm to 80% at 1400 nm. Therefore, the DW factor is overestimated by less than 2%. The maximum reported DW factor is 16% and 18% for a single G center and ensemble, respectively. Our value is comparable with the DW factor of the G center ensemble, for which the annealing protocol is optimized. We have changed the text accordingly.

Reviewer #3

“The paper claims that they could prove fabrication of single photon emitters compatible with CMOS technology. The topic is indeed a potential subject for Nature Communications.”

We are glad to know that the topic of our manuscript is a potential subject for Nature Communications.

“The main problem of the paper is about the goodness of the measurement of the single photon emitters that is the key focus of the entire paper. In the paper, the single photon nature of the emission is defined as $g^{(2)}(0) < 0.5$, however, it has been shown recently that this definition is misleading because, in practice, $g^{(2)}(0) < 0.5$ condition does not absolutely guarantee single photon emission because two-three emitters may also produce $g^{(2)}(0) \sim 0.4$, for instance. Recent discussions can be found in Refs. arXiv:2111.01252 and arXiv:2203.11859. This issue should be resolved tranquillizingly before considering for recommendation for publication this paper in any form.”

The Reviewer points out an important issue. According to the procedure described in arXiv:2111.01252, the single photon emission is unambiguously confirmed when $g^{(2)}(0)$ is zero

after background and time jitter corrections. In fact, we have performed this analysis. Equations (21), (22) and (25) in arXiv:2111.01252 are equations (2), (4) and (1) in our manuscript. The correction due to time jitter (40 ps for the detectors and 14 ps for the time tagger) is negligible, as it is by more than two orders of magnitude shorter than the τ_1 time in Eq. (1). Using background correction, we obtain $g^{(2)}(0) = 0$ as shown in Fig. S3b in the Supplemental Material.

In the revised version of our manuscript, we present corrected $g^{(2)}$ functions for all second-order autocorrelation measurements in Figs. 1c, 2d, 4b and 5b. All these $g_{\text{corr}}(\tau)$ functions approach zero at $\tau = 0$, pointing at single photon emitters. We have added a paragraph describing this procedure and cite now arXiv:2111.01252 and arXiv:2203.11859.

REVIEWERS' COMMENTS

Reviewer #1 (Remarks to the Author):

The Authors have addressed all my concerns. I particularly appreciate the additional experiment they did in response to my comment #1 (new fig. 5), which nicely extends the range of techniques for controlled creation of silicon colour centres. I recommend publication as is.

Reviewer #3 (Remarks to the Author):

The authors replied to the comment of mine and other comments satisfactorily. It is assumed that the background correction was carried for each spot to carry out the statistical analysis. The paper is recommended for publication to Nature Communication.